**www.cambridge.org/ext**

biodiversity loss; climate change; distribution; habitat loss; scenarios and forecasts

**Author for correspondence:**
Damaris Zurell,
Email: damaris@zurell.de

# Predicting extinctions with species distribution models

Damaris Zurell[1] ⓘ, Susanne A. Fritz[2,3] ⓘ, Anna Rönnfeldt[1] and Manuel J. Steinbauer[4,5,6] ⓘ

[1]Institute for Biochemistry and Biology, University of Potsdam, Potsdam, Germany; [2]Senckenberg Biodiversity and Climate Research Centre (S-BiKF), Frankfurt, Germany; [3]Institut für Geowissenschaften, Goethe University Frankfurt, Frankfurt, Germany; [4]Bayreuth Center of Ecology and Environmental Research (BayCEER), University of Bayreuth, Bayreuth, Germany; [5]Bayreuth Center of Sport Science, University of Bayreuth, Bayreuth, Germany and [6]Department of Biological Sciences, University of Bergen, Bergen, Norway

## Abstract

Predictions of species-level extinction risk from climate change are mostly based on species distribution models (SDMs). Reviewing the literature, we summarise why the translation of SDM results to extinction risk is conceptually and methodologically challenged and why critical SDM assumptions are unlikely to be met under climate change. Published SDM-derived extinction estimates are based on a positive relationship between range size decline and extinction risk, which empirically is not well understood. Importantly, the classification criteria used by the IUCN Red List of Threatened Species were not meant for this purpose and are often misused. Future predictive studies would profit considerably from a better understanding of the extinction risk–range decline relationship, particularly regarding the persistence and non-random distribution of the few last individuals in dwindling populations. Nevertheless, in the face of the ongoing climate and biodiversity crises, there is a high demand for predictions of future extinction risks. Despite prevailing challenges, we agree that SDMs currently provide the most accessible method to assess climate-related extinction risk across multiple species. We summarise current good practice in how SDMs can serve to classify species into IUCN extinction risk categories and predict whether a species is likely to become threatened under future climate. However, the uncertainties associated with translating predicted range declines into quantitative extinction risk need to be adequately communicated and extinction predictions should only be attempted with carefully conducted SDMs that openly communicate the limitations and uncertainty.

## Impact statement

Extinction is the irreversible loss of unique life forms. Ongoing climate change is predicted to cause significant loss of biodiversity, meaning loss of species, genes and ecosystems. This could lead to multiple negative consequences for human society as important ecosystem functions are also being lost. Understanding and predicting species extinctions for scenarios of future climate change is thus of main interest for science and people. Most estimates of future extinction risk rely on correlative species distribution models (SDMs). These relate the observed distribution of the focal species to observed environmental characteristics and then make forecasts where the species will find suitable environmental conditions in the future. We summarise how these models can be used to predict extinctions and what are the challenges and limitations of this approach. For example, these models ignore how long it might take until species go extinct after the loss of their habitat. Many processes and factors determining the loss of the few last individuals of a species are currently not well understood, and we highlight where particular care must be taken in the model building steps and where more detailed investigations into these processes are needed to improve predictions of species extinction risks. Despite prevailing challenges, there is high demand for estimates of future extinction risk. Overall, SDMs currently provide the most accessible method to estimate climate-related extinctions across multiple species, and those predictions, although uncertain, are needed by society to prepare adaptive strategies and policies for mitigating the consequences of human-induced climate change.

## Introduction

Species distribution models (SDMs) are the main source to estimate the magnitude of climate change-related species extinctions (Urban, 2015; Warren et al., 2018), although inferring extinction risks from SDMs is controversial (Hampe, 2004; Dormann, 2007; Araújo and

Peterson, 2012). Currently, SDMs are the most widely used tools for assessing climate change impacts on biodiversity (Araújo et al., 2019) and for evaluating current and future ranges of species (e.g., Thomas et al., 2004). Many SDM studies have attempted forecasts under climate scenarios, for example, assessing climate change vulnerability of species in terms of potential range loss (e.g., Zhang et al., 2015; Martín-Vélez and Abellán, 2022), differences in seasonal range loss for species with different IUCN Red List status (Zurell et al., 2018) or contrasting effects of dispersal or local adaptation on climate-related extinction risk (Thuiller et al., 2006; Román-Palacios and Wiens, 2020). Also, SDM-derived extinction risk estimates regularly inform political processes (IPCC, 2022, Chapter 2.5).

Despite the widespread use of SDMs in climate change research, they also remain criticised and regularly spark debate (Dormann et al., 2012; Thuiller et al., 2013). For example, when Thomas et al. (2004) inferred global estimates of extinction risks across plants and animals by synthesising studies that applied SDMs under future climate change scenarios, their results were criticised in several commentaries (Buckley and Roughgarden, 2004; Thuiller et al., 2004). Main criticisms referred to conceptual challenges related to how SDM predictions relate (or not) to extinction risk of species, to underlying SDM assumptions and to methodological challenges. Still, the number of SDM applications, also in relation to global change, is constantly increasing (Araújo et al., 2019) and meta-analyses indicated that of all studies that estimate extinction risk under future climate, 76% are based on SDMs (Urban, 2015). In the absence of better-suited alternative methods to estimate climate change-related extinction risk, SDMs seem to remain the most practical methodology despite well-founded criticism. In this review, we first explain the basics of SDMs and provide a literature overview over the use of SDMs for quantifying climate change impacts and extinction risk. Then, we describe how SDMs are currently used to inform extinction risk estimates and discuss the contentions from conceptual and methodological viewpoints. Finally, we summarise current good practice.

## What are species distribution models?

Correlative SDMs (a.k.a. habitat suitability model, ecological niche model and environmental envelope model, among others; Elith and Leathwick, 2009) relate geographic occurrences of organisms to prevailing environmental conditions (abiotic, biotic or both) within a statistical or machine-learning framework (Guisan and Zimmermann, 2000; Guisan and Thuiller, 2005). The inferred model describes the species–environment relationship informing how habitat suitability scales with different environmental predictors. This relationship can be projected into geographic space using layers of environmental predictors to predict suitable habitat under current environment or scenarios of future (or past) environments (Figure 1). In many cases, SDMs are fit to climatic predictors, especially when predicting future scenarios. When prediction is the goal, predictive accuracy and transferability to new times and places should be validated, which is not trivial as independent data are often missing (Araújo et al., 2005; Yates et al., 2018; Zurell et al., 2020). By applying a threshold approach (Liu et al., 2005, 2013), the model output of habitat suitability can be transformed into predicted presence (and predicted absence), which can be interpreted as potential distribution of the species given the environmental conditions. The realised distribution of the species might deviate from the predicted potential distribution because of underlying ecological processes and methodological challenges (Figure 1; Soberón, 2007; Elith and Leathwick, 2009), which we will further discuss below.

Recent years have seen considerable advances in SDM algorithms and accompanying methods for fitting SDMs (Valavi et al., 2021). Also, digital availability of biodiversity data and environmental data including climate scenarios has increased strongly (Wüest et al., 2020). These factors have contributed to widespread

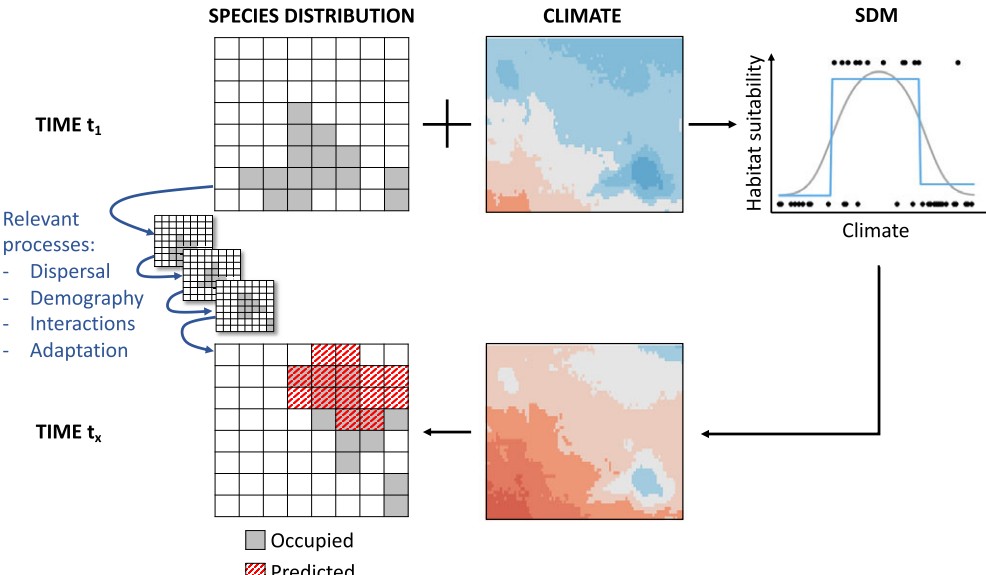

**Figure 1.** Conceptual overview of correlative species distribution models (SDMs) used for prediction under climate change. SDMs are fitted to observed occurrence data and climatic (or, more generally, environmental) data in time step $t_1$ (upper row of figures) using adequate statistical and machine-learning approaches (top-right plot shows two example approaches as grey curve and blue step function). The fitted species–environment relationship is then used to make predictions of habitat suitability and potential distribution at time step $t_x$ given future climate (or environmental) layers (lower row of figures). The potential future distribution derived from SDMs can differ from the true distribution at time step $t_x$ as the latter will be co-determined by the biological processes of dispersal, demography, species interactions and genetic or behavioural adaptation leading to transient dynamics (small figures in the middle).

**Table 1.** Web of Science search terms used in the literature search on 21 July 2022

| Topic | Search terms |
|---|---|
| Species distribution model | TS = ("species distribut*" OR "habitat distribut*" OR "climat* envelope" OR bioclimat* OR "habitat suitab*" OR niche OR "resource selection" OR SDM OR ENM OR BEM OR BCM OR HSM OR RSF) AND (model*) |
| Climate change | TS = ("climat* change") |
| Extinction | TS = ("extinct*" OR "pop* declin*") |
| Time period | PY = (1900–2021) |
| Document types | Article |
| Languages | English |
| Web of Science index | Science Citation Index Expanded (SCI-EXPANDED) – 1900 to the present |

use of SDM techniques. To quantify how many SDM-related studies exist and how many of these target climate change and species extinctions, we conducted a keyword-based literature search in the Web of Science on 21 July 2022 for papers published in 1900–2021 (see Table 1 for list of keywords). First, we identified all studies that mention SDMs (or synonyms). This revealed more than 40,000 studies published over all disciplines, with the first SDM mention in 1969 and a steady increase since 1990 (Figure 2). We then refined the list of papers by adding keywords related to climate change (Table 1). Of all SDM studies, climate change was mentioned in c. 20% and with increasing frequency through time (Figure 2C). Finally, we further refined the list of papers by adding keywords related to extinction or population declines (Table 1). Interestingly, extinction was mentioned only in one study related to SDMs and climate change before 2002. Since then, the absolute number of climate change-related SDM studies mentioning extinction increased, culminating in 171 such studies published in 2021. Yet their relative proportion decreased over time (average proportion c. 18%; Figure 2C).

As this simple keyword search could provide an overoptimistic number of hits, we assessed a randomly drawn subset of 300 publications from the final set of articles that mentioned SDMs, climate change and extinction in more detail (see the Supplementary Material). All articles were screened by the same assessor, first screening the abstracts for determining whether the study applied SDMs and then screening the entire article for inclusion of climate change scenarios and quantitative estimates of extinction risks. Of the 300 articles, 203 publications indeed used SDMs, 161 applied SDMs under climate scenarios (150 under future scenarios and 11 under historic scenarios), 134 quantified future climate-related range changes from SDMs and 74 of these studies implied or provided inference on extinction risk (see the Supplementary Material). Thus, while the large majority of SDM studies do not explicitly aim at predicting extinctions, SDMs are clearly used for deriving species' extinction risk under climate change. In fact, most extinction risk estimates reported in the IPCC AR6 are based on SDMs (IPCC, 2022, pp. 256–261).

## How is extinction risk derived from species distribution models?

SDMs can predict climatically suitable areas (or, more generally, environmentally suitable areas). An assumed increase in extinction risk with the decline in suitable habitat underlies most estimates of extinction risks from SDMs. They reflect theoretical understanding from island biogeography that smaller areas can harbour less individuals and smaller populations face a higher risk of extinction (species–area relationship [SAR]; MacArthur and Wilson, 1967). While generally accepted, the precise relationships between range size decline, population decline and extinction probability are unknown for most species (reviewed in Mace et al., 2008). Often, guidelines from the extinction risk classification formalised by the IUCN Red List of Threatened Species (IUCN, 2001, 2022) are used for translating SDM-derived estimates of range-size declines to extinction probabilities (e.g., Ahmadi et al., 2019; IPCC, 2022), while this simplified translation lacks empirical evidence (Akçakaya et al., 2006). Originally, the IUCN "categories of threat […] provide an assessment of the likelihood that […] the species will go extinct within a given period of time" (Mace and Lande, 1991). Based on quantitative analyses such as a population viability analysis (PVA), species are, for example, classified as "critically endangered" when they face a >50% likelihood of extinction within the coming 10–100 years (depending on generation times), and "endangered" with a likelihood of extinction of >20% (criterion E; IUCN, 2001, 2022). However, due to insufficient or uncertain training data for PVA models, extinction probability estimates based on quantitative population viability analyses are missing for most species.

As alternative to quantitative analysis (criterion E), simpler estimates of species range decline can be used in the IUCN framework, for example, to classify a species as "critically endangered" or "endangered" if it is predicted to lose ≥80% or ≥50% of its range, respectively, over the longer of 10 years or three generations (subcriteria A3 and A4). Criterion A was devised for observed population decline, but it is now also applied to SDM-derived estimates of future range size declines (Mace et al., 2008; IUCN, 2022). However, it is important to note that while the IUCN allows using a future decline of range size (subcriteria A3 and A4) or an extinction probability estimate (criterion E) for classifying species into the same extinction risk category with well-funded arguments, this does not mean that a certain decline in range size can be translated into a specific quantitative extinction risk (Akçakaya et al., 2006; Mace et al., 2008). Accordingly, the IUCN Red List guidelines state that "the risk-based thresholds of criterion E should not be used to infer an extinction risk for a taxon assessed […] under any of the criteria A to D″ (IUCN, 2022, p. 62). To illustrate this, a projected range loss of ≥80% may be used to classify a species as "critically endangered" (according to subcriteria A3 and A4), but this does not mean that its probability to go extinct within three generations is larger than 50% only because the latter would also be a valid criterion for being classified as "critically endangered" (according to criterion E). Yet such a use of SDM-based range changes paired with IUCN criteria for extinction risk assessment is also – misleadingly – stated in the latest IPCC report (IPCC, 2022, p. 257), where central quantifications of extinction risks in IPCC AR6 are based on this approach (Warren et al., 2018). Here, we want to echo Akçakaya et al. (2006) and Mace et al. (2008) and caution against such interpretation.

When using SDMs to project extinctions, a better understanding of the relationship between range-size or abundance declines and extinction risk is thus central. Recent meta-analyses came to mixed conclusions on whether habitat suitability is a reasonable proxy of abundance (Weber et al., 2017; Lee-Yaw et al., 2022). This relationship of suitable habitat and extinction risk is likely dependent on species-specific characteristics (e.g., life-history strategy and

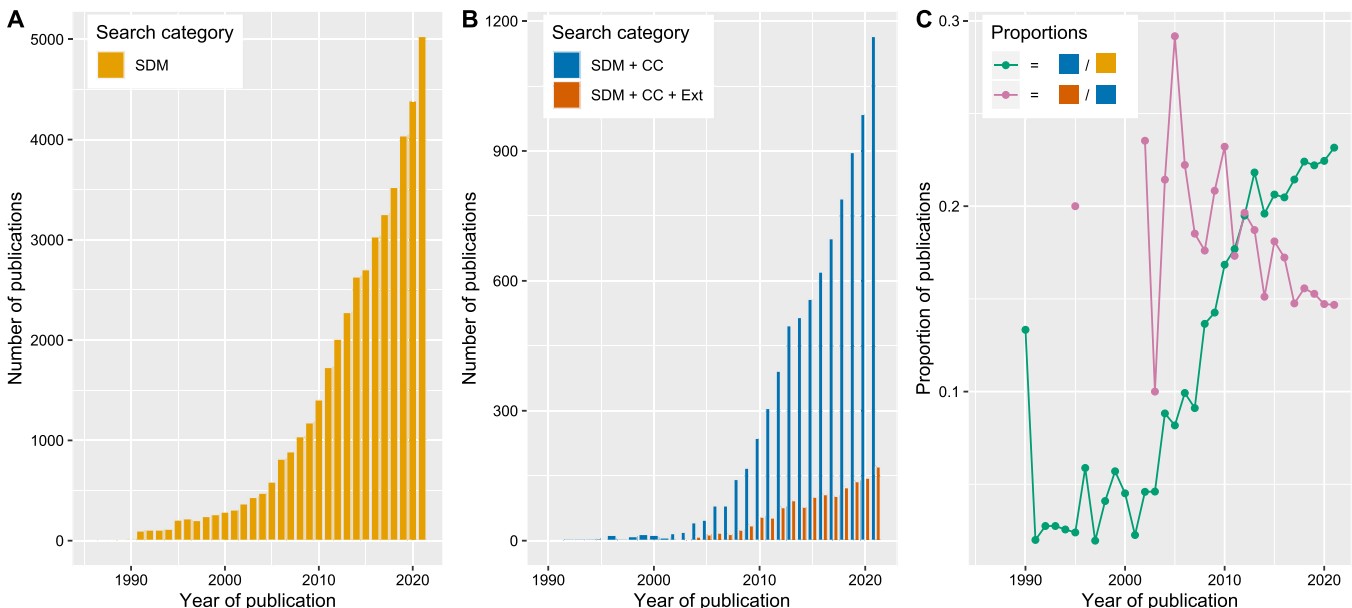

**Figure 2.** Use of correlative species distribution models (SDMs) over the last three decades. We extracted all studies from the Web of Science (see the keywords in Table 1) between 1900 and 2021 and classified them according to whether they were used in a climate change context and whether they mentioned extinctions or population declines. Earliest SDM studies appeared in 1969 with one to three publications per year until 1985. For easier visualisation, we only show publications published after 1985. (A) shows the absolute number of SDM publications per year. (B) shows the absolute number of SDM publications that mention climate change (CC) and those that mention both CC and extinctions (Ext). (C) shows the proportion of different SDM studies per year: green indicates the proportion of all SDM studies per year that mention climate change and purple indicates the proportion of all climate change-related SDM studies per year that mention extinction or population decline.

generation times). Most importantly, the relationship between population size and habitat (suitable area) may not be linear (Blackburn et al., 2006). In fact, SDMs only predict the potential distribution, while demographic and ecological processes and random events may prevent the species from occupying all suitable habitat (Figure 1). Here, the debate on whether SDMs capture fundamental or realised niches is relevant (e.g., Soberón, 2007; Holt, 2009).

On a conceptual level, the process of extinction is very hard to quantify with any method. An alternative method to predict extinction under future climate change relies on SARs, that is, on strong empirical evidence for ubiquitous relationships between species richness and geographical area size (Matthews et al., 2021). These SARs can be utilised to predict changes in species richness given projected changes in area, for example, in the geographic extent of a given habitat under climate change (Pimm et al., 1995). However, this approach has been criticised because empirical SARs depend on species and environmental characteristics (Matias et al., 2014; Schrader et al., 2020), because extinction may often lag substantially behind habitat loss (Triantis et al., 2010), but most importantly because methods to construct a SAR are unable to adequately integrate the distribution of last individuals and to differentiate underlying sampling problems from the actual loss of these last individuals (He and Hubbell, 2011, 2013; Kitzes and Harte, 2014). Further, the method is not species-specific but relies on defining relevant areas where habitat will be lost, as the amount of lost area is used to predict species richness (rather than extinction probability for individual species). SARs are thus better suited to deal with land use-related habitat loss rather than climate change-related extinction, and they cannot be used to inform about the risk of single species that would be relevant for conservation.

In summary, SDMs currently provide the most workable, species-specific prediction tool for threat classification under climate change, but should be used with caution. The translation of a

decline in range size as projected by SDMs into quantitative extinction risk estimates (as by the IUCN criterion E) is fraught with difficulty, because the basic underlying extinction–range decline relationship is unclear, and likely to differ among taxa and environments. However, in the absence of more appropriate methods, careful application of IUCN criteria to SDM projections allows categorising species-level extinction risk from future climate change under specific assumptions. In the following two sections, we review conceptual and methodological challenges of SDMs that are particularly relevant to this process.

## Conceptual challenges

Using SDMs for making predictions about future extinctions hinges on the expectation that these models make reliable predictions into the future. There are several reasons why this is not necessarily true. SDMs make several critical assumptions when applied to global change scenarios, most importantly that species are in equilibrium with current environment and will achieve (instantaneous) new equilibrium in the future, and that all environmental constraints are adequately understood and considered in the model (Elith and Leathwick, 2009; Zurell et al., 2020). Here, we discuss why these critical assumptions are unlikely to be met in many cases. In addition, SDMs assume that species will conserve their niches into the future, for example, that no change in the species–environment relationship will occur through adaptive evolution of thermal tolerance, which is probably unlikely given strong selective pressure (Dawson et al., 2011; Buckley and Kingsolver, 2012).

There is increasing evidence that range-shifting species are lagging behind their climatically suitable habitat (Svenning et al., 2008), leading to suitable habitat not yet colonised ("colonisation credit") and indicating departure from the equilibrium assumption underlying SDMs. Impacts from other global change drivers such as

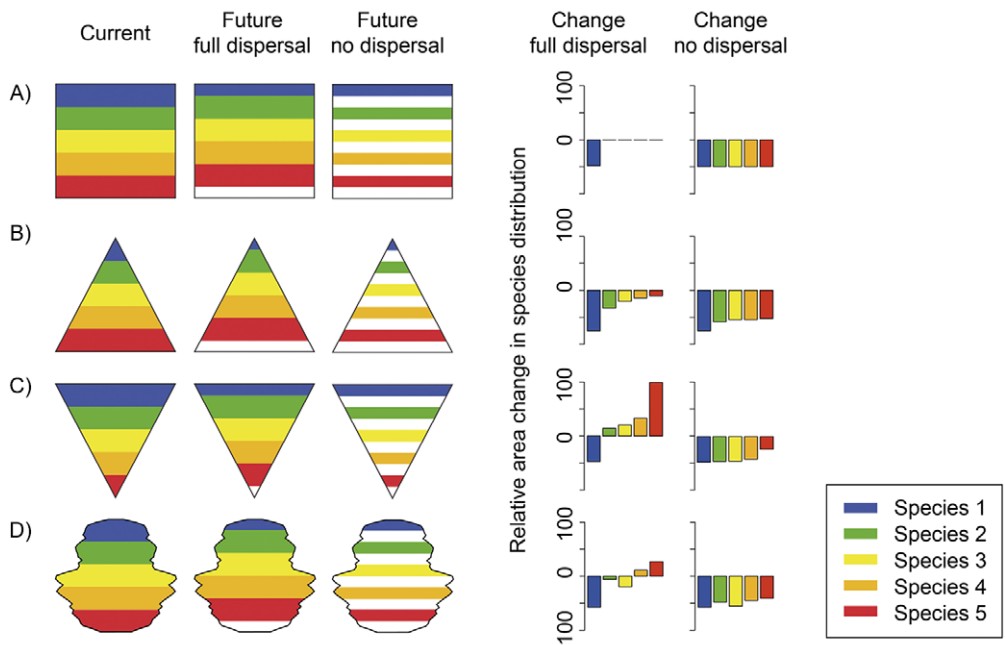

**Figure 3.** Shape of the study area as well as dispersal assumptions influence predictions of correlative species distribution models (SDMs). This is shown here for theoretical continents characterised solely by a linear gradual decrease of temperature to the upper part of the study area. We assume that each temperature band is occupied by one hypothetical species. In the future, temperature isoclines will move upwards on the shown study areas (imitating global warming; sketch maps on the left). Under the full-dispersal assumption, species will fully track their suitable temperature band. Under the no-dispersal scenario, species will lose climatically suitable area but will not shift their range. These two extremes reflect the most common dispersal assumptions in SDM-based projections under climate change. Extinction risk estimates derived from SDMs strongly depend on the geographical shape of the study area, and the dispersal assumption (bar charts on the right showing relative area change for each species). Fun fact: the continent map in (D) is a rough representation of the area–latitude relationship of western Europe.

habitat destruction and land fragmentation can prevent a species from tracking suitable climate, leading to a much higher extinction risk than can be suggested by an SDM (Travis, 2003; Hof et al., 2011). The dispersal ability of a species will (co-)determine the climate tracking ability, yet reliable empirical estimates are largely missing (Bullock et al., 2017; Fandos et al., 2023). When comparing potential current and potential future distribution, SDMs often assume full dispersal or no dispersal (Thuiller et al., 2019). In the first case, we assume that species fully track the changing climate in space. In the second case, we assume that species are not at all shifting their range but simply lose currently suitable climate area. Inference of range size declines can dramatically differ between these extreme assumptions and are also strongly influenced by the geography of the study area (Figure 3). Additionally, climate tracking and range shifting can be affected by demographic processes, adaptive evolution and species interactions (Buckley and Kingsolver, 2012; Svenning et al., 2014; IPBES, 2016; Schleuning et al., 2020). For example, the presence of competitors, generalist consumers or predators can slow down range expansion (Davis et al., 1998). At the same time, long life expectancy of species can result in (temporary) survival under unfavourable conditions and delayed local extirpations (extinction debts; Kuussaari et al., 2009). To some extent, these violations of the equilibrium assumption can be captured by the two extremes of full versus no dispersal scenarios. Yet it is highly uncertain towards which of these extreme assumptions a specific species will lean. Therefore, the IUCN (2022) recommends to derive and overlap future SDM predictions at one-generation intervals to assess climate-tracking potential. Ideally, this should be coupled with reasonable assumptions about potential spread of species in the face of the above-mentioned processes.

It is almost impossible to incorporate all relevant environmental constraints into SDMs or project these into the future, even though this is a core assumption underlying SDM methodology. Ecological processes are highly scale-dependent. Climatic conditions may govern the broad-scale species distribution, while the fine-scale distribution may be determined by local resources (Guisan and Thuiller, 2005) or microclimate (Suggitt et al., 2011). Consequently, climate niche tracking and range shifting may ultimately be limited by fine-scale resource distributions (Skov and Svenning, 2004; Dormann, 2007; Suggitt et al., 2018). Particularly among plants, facilitative effects of other species may strongly influence habitat suitability under unfavourable conditions (like nurse plants in alpine or dry environments; Steinbauer et al., 2016; Gallien et al., 2018). Not considering these fine-scale environmental or biotic predictors in SDMs may bias predictions, and lead to under- or over-estimation of suitable habitat. Additionally, predictions into the future require availability of environmental scenarios. Climate models are well advanced, and the climate science community produces regular updates on climate scenarios for the IPCC (Knutti et al., 2013). For land use, which is another major determinant of species distribution, future scenarios are less well developed and more uncertain due to unknown political and economic development (Cabral et al., 2022).

A further challenge arising with future predictions is unidentified constraints in species distributions. Environmental or land use factors that constrain the distribution of a species can only become apparent as effective predictors in SDMs if they limit the current distribution of a species. If, for instance, soil characteristics are largely suitable within the current range of a species that is currently constrained by climatic factors, an SDM will not identify soil as a relevant predictor variable and will be unable to identify that a

projected future range may be largely uninhabitable due to unsuitable soil conditions. Particularly, edaphic variation is a major determinant of plant distributions (Hulshof and Spasojevic, 2020) but is often neglected in SDMs, possibly because soil characteristics on macroscales correlate with climate (e.g., along latitude; Huston, 2012). In fact, the identification of relevant variables only based on explanatory power may be very misleading, as random spatial variables may be able to predict spatial distribution pattern as well as commonly used environmental predictors (Fourcade et al., 2018). As SDMs are phenomenological models, it is important not to mistake correlation for causation (Dormann et al., 2012). Also, phenomenological relationships might not hold in the future if species adapt to novel abiotic and biotic conditions.

## Methodological challenges

When using models to estimate extinction risks and to inform management and provide policy support, it is important that models are fit for this particular purpose. This may be challenged by conceptual problems, as discussed above, and by methodological challenges, for which we give a brief overview here. Although SDMs are commonly perceived as a simple method, only few studies achieve quality standards that will match the standards specified by the IUCN for extinction risk assessment (IUCN, 2022). Several recent papers provide guidance and propose best-practice standards for ensuring SDM credibility for decision-making and biodiversity assessment (Araújo et al., 2019; Sofaer et al., 2019). Nogués-Bravo (2009) discussed how SDMs can be used to predict past distributions of species' climate niches and derived a set of recommended practices for hindcasting, arguing that inadequate methods can lead to "a cascade of errors and naïve ecological and

evolutionary inferences" (Nogués-Bravo, 2009). Although focused on hindcasts, the identified methodological challenges also apply to forecasting SDMs as a basis for estimating extinction risk. We summarise these below as (1) model specification, (2) selection of environmental predictors, (3) model validation and (4) uncertainty through non-analogue climates (Barry and Elith, 2006; Nogués-Bravo, 2009; IUCN, 2022; Figure 4A).

(1) Several studies have shown that algorithmic choices can strongly affect current and future range predictions (Buisson et al., 2010; Thuiller et al., 2019). Algorithms range from simple profile or envelope methods, regression-based approaches to complex machine-learning methods (Guisan et al., 2017). Machine-learning methods derive complex species–environment relationships that closely fit the observed data and have often been reported to achieve highest prediction accuracy (Elith et al., 2006; Valavi et al., 2021). Yet overfitting might also lead to reduced transferability to new times and places, and simpler models might thus be preferable for predicting future species ranges and extinction risk (Merow et al., 2014; Brun et al., 2020). For making biodiversity predictions under scenarios of climate change, the IUCN advises to use at least three different SDM algorithms of intermediate complexity to capture the uncertainty related to species–environment relationships (IUCN, 2022). (2) Similarly, the number of predictor variables included in a model should be kept reasonably small when predicting into the future (Brun et al., 2020). Also, uncertainty in available environmental data should be considered, for example, when alternative data sources are available. As SDMs are increasingly used in global change research, the question of transferability also becomes more urgent (Yates et al., 2018). (3) Due to a lack of independent test data, validation is typically done based on data partitioning (Araújo et al., 2005). Newest

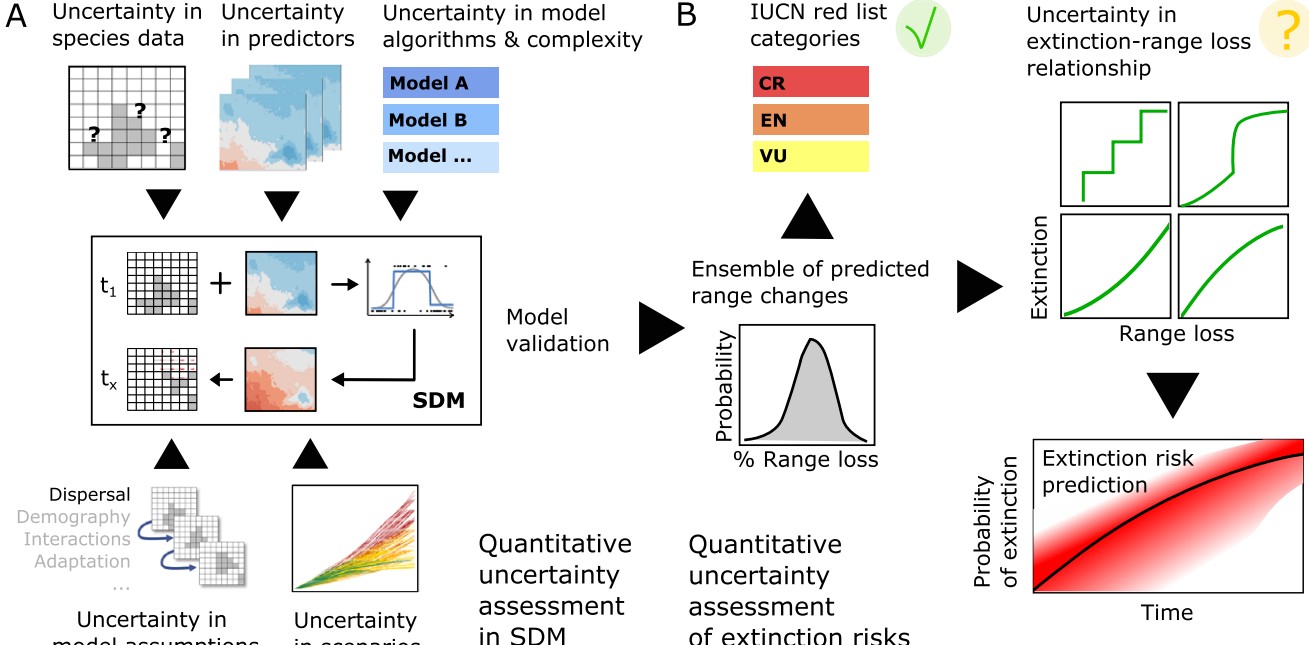

**Figure 4.** Workflow and challenges for deriving adequate range loss predictions from correlative species distribution models (SDMs) and subsequent estimates of extinction risk. (A) Several methodological and conceptual challenges should be considered in SDM development, and resulting uncertainty should be adequately communicated. Current best practices for achieving or assessing model credibility are summarised in Araújo et al. (2019) and Sofaer et al. (2019). (B) While predicted range loss can be readily translated into IUCN Red List categories for threatened species following the IUCN Red List guidelines (IUCN, 2022), the IUCN advices against deriving quantitative extinction risk estimates from SDM predictions. At the very least, further research is required regarding adequate extinction–range loss relationships and adequate uncertainty propagation (IUCN Red List categories: CR, critically endangered; EN, endangered; VU, vulnerable).

developments in model validation now advocate for spatial or environmental block cross-validation approaches that strategically hold out data that are spatially or environmentally clustered, and by this force extrapolation during validation (Bagchi et al., 2013; Roberts et al., 2017; Valavi et al., 2019). Although an important step forward for assessing prediction uncertainty, we are still missing clear guidance about adequate block design to ensure robust estimates of prediction accuracy of future ranges. Validation of prediction accuracy into the future is further complicated by the fact that global change could lead to non-stationarity in the processes that govern the inferred species–environmental relationship (Rollinson et al., 2021), potentially violating the assumption of niche constancy. (4) Lastly, when making predictions to the future, we also need to consider uncertainty through extrapolating to novel environments. Different algorithms will exhibit different extrapolation behaviour and it is thus advisable to explicitly assess environmental novelty (Elith et al., 2010; Zurell et al., 2012). The IUCN (2022) lists several of these methodological issues that need to be considered and communicated to include SDM results and predictions in Red List assessments, and we highly recommend consulting these guidelines when planning studies estimating extinction risk (Figure 4A). Best practices for achieving or assessing model credibility are also summarised in Araújo et al. (2019) and Sofaer et al. (2019).

### Moving forward: Predicting uncertainty is better than wrong predictions

While the outlined conceptual and methodological challenges illustrate why using SDMs for predicting extinction risk is potentially problematic, SDMs are still the most widely applicable tool currently available for predicting species' potential future distributions under climate change. Our survey of SDM-related challenges also highlights crucial future research questions that need to be addressed to improve the use of SDMs for predictions of extinction risk from climate change (Figure 4B). In particular, improved understanding of the relationship between species' extinction probability and (SDM-derived) range decline is of central importance for more robust predictions. Particularly, the non-random distribution and extinction probability of species that have declined to very small population sizes are not well understood and pose the largest uncertainty when estimating extinction probabilities. In the absence of clearly identifiable climate-related extinctions over the last centuries, science cannot build on empirical evidence when assessing climate change-related extinction risk, at least not in the recent past. In fact, until now, only two global species extinctions in modern times can be attributed with confidence to human-induced climate change (IPCC, 2022, p. 237). This absence of observed climate-related extinctions does not contrast with model predictions, but it limits our current understanding of extinction processes and constraints testing the precision of predictive models (Brook et al., 2008). The most promising ways forward for a better understanding of the extinction–range loss relationship may thus be the investigation of local extirpation patterns, of spatially explicit simulations, and, above all, of the rich information on past biotic responses to climate changes provided by the fossil record (Calosi et al., 2019; Fordham et al., 2020).

A second area of future research relates back to critical methodological issues of SDMs. In many studies, species with very small range size (or a low number of occurrences) are excluded, because SDMs need a certain number of data points, yet these highly endemic species are among the most relevant for conservation (Lomba et al., 2010; Breiner et al., 2018). As rarity could have several reasons and relate, for example, to a narrow niche or to climatic rarity (Ohlemüller, 2011), the extinction risk–range loss relationship might even be different for species that have evolved to occur in small areas or at low populations size compared with species that are forced to do so. Also, little consensus exists yet regarding adequate assumptions for considering potential future spread in SDM predictions. Many researchers have called for integrating more process detail into distribution models to account for relevant transient dynamics under global change (Figure 1; IPBES, 2016). Yet, until such process-based models are available for large numbers of species, an intermediate solution could be to agree on standards for incorporating reasonable assumptions about species spread (mediated by dispersal, demography, and species interactions, among other processes) into predictions of range changes.

In the face of the climate and biodiversity crises, there is a clear demand of future extinction risk estimates (IPCC, 2022). Thus, while advancing on central research questions related to improve our fundamental understanding of extinction processes, we advocate that well-conducted SDMs should initially fill the knowledge gap and make predictions on extinction risk – but only when following good practice and when openly communicating the limitations and uncertainty (Araújo et al., 2019; Feng et al., 2019; Zurell et al., 2020; IUCN, 2022). Particularly, SDMs used for estimating future extinction risk must be used with caution and constructed with care, and the application of IUCN Red List criteria to SDM results must follow the published guidelines. According to these, predicted range size declines from well-constructed SDMs can readily be used to classify a species as threatened from climate change. In contrast, it is seen as problematic to back-infer quantitative extinction risks from this classification, yet this is currently being done in policy-relevant reports such as IPCC (2022). We thus suggest as middle ground that the uncertainties associated with translating range declines into quantitative extinctions risks should be more adequately communicated while at the same time increasing research efforts to better understand the extinction risk–range decline relationship (Figure 4). In addition, we need to acknowledge that SDMs will only provide predictions of suitable area while ignoring other relevant processes affecting range shifts and extinction. It is thus important that we also assess uncertainty in model predictions induced by alternative assumptions about the climate-tracking potential of species, for example, through phenotypic plasticity, local genetic adaptation or variability in dispersal. We follow recent publications in arguing that the SDM community is largely aware of these issues and has developed improved standards (Araújo et al., 2019; Feng et al., 2019; Sofaer et al., 2019; Zurell et al., 2020) that will allow future studies to address methodological issues and handle conceptual issues carefully.

**Open peer review.** To view the open peer review materials for this article, please visit http://doi.org/10.1017/ext.2023.5.

**Supplementary materials.** To view supplementary material for this article, please visit http://doi.org/10.1017/ext.2023.5.

**Data availability statement.** The review table of the 300 randomly selected papers is available in the Supplementary Material.

**Author contributions.** Conceptualisation (equal): D.Z., S.A.F., M.J.S.; Investigation (equal): A.R.; Investigation (lead): D.Z.; Methodology (equal): D.Z., S.A.F., M.J.S.; Methodology (support): A.R.; Supervision (lead): D.Z.; Visualisation (equal): M.J.S.; Visualisation (lead): D.Z.; Writing – original draft

preparation (equal): S.A.F., M.J.S.; Writing – original draft preparation (lead): D.Z.; Writing – review and editing (equal): D.Z., S.A.F., M.J.S.; Writing – review and editing (support): A.R.

**Financial support.** The work was supported by the Deutsche Forschungsgemeinschaft DFG (D.Z., grant ZU 361/1-1; M.J.S., grant STE 2360/2-1 embedded in the Research Unit TERSANE FOR 2332), the Leibniz Association (S.A.F., grant number Leibniz Competition P52/2017) and the ERC H2020 research and innovation programme (M.J.S., project HOPE grant 741413).

**Competing interest.** The authors declare none.

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
