## [Reviewer Report]

*Comments to Author*: The paper by Zurell et al. provides a very balanced review of the benefits and drawbacks of using SDMs for predicting extinctions for species. Overall, I found the paper useful and informative. I have two major suggestions that the authors should consider as they revise their manuscript. It would be helpful if the paper could explicitly summarize: 1) best practices and 2) future areas of research that are most critical. The authors could put such sections into the text but another option would be to create some sort of table or flow chart for each of these. I believe succinct summaries of these two items would be most useful to potential readers.

Specific comments:

-Line 81: although inferring extinction risks from SDM is controversial

-Line 83: remove certainly

-Figure 1 should be remade so that it is not hand drawn. The current sketch looks like it might be a place holder.

-Lines 139-158: there is no information how this literature review was conducted or how these articles were found. Did the authors follow the PRISMA guidelines for systematic review (http://www.prisma-statement.org/)? Please add these methods to the text or at least an appendix. There is a brief statement of the methods in the Figure 2 legend but this should be clarified.

-Line 334: sentence is awkward. Please rephrase- it may just be typos.

- Line 338: what is the definition of “assessment criteria” in the context of SDMs? Please provide a definition.

-Lines 359-372: Suggest noting that an important part of any prediction is the notion of temporal stationarity in covariate estimates, which is not guaranteed. I realize that the problems of nonstationarity are prevalent and extend beyond SDMs but it’s an important limitation (Rollinson et al., 2021, Frontiers in Ecology and the Environment)

-Line 376: Not sure that SDMs are the best tool. There are many other, more mechanistic based approaches that are probably better. But SDMs are a pretty general tool that can be used in situations where there might not be a lot of available data (or the data are pretty easy to collect, i.e., opportunistic presence-only) and across lots of species. So the method is maybe the most popular but is it the best? Consider making that distinction.

-Line 412: This is an important point that may be missed. Consider adding another sentence or two explain why this is.

---

## [Editor Report]

*Comments to Author*: Dear Dr. Zurell and colleagues

Thank you for submitting your manuscript to Extinction. I have received one review from a trusted source who has expertise in both the development and application of SDMs. Because this review is sufficiently positive, I am recommending that you undertake 'minor revisions' and submit an updated version of the manuscript. 

Please be sure to attend to all of the reviewer's comments, especially the part about the need for brief summaries in two areas.

---

## [Reviewer Report]

*Comments to Author*: Thanks to the authors for their efforts in revising their manuscript in response to the previous comments. The new sections on best practices, key uncertainties, and future areas of research should be useful to potential readers and in guiding the field forward. The revised figure 1 and the new figure 4 do a nice job of communicating the main concepts.

---

## [Editor Report]

*Comments to Author*: Dear Dr. Zurrell

I have reviewed the revisions provided by the authors (EXT-22-0019.R1) in response to the earlier editorial decision of 'Minor Revisions'.

The reviewer is satisfied with your changes and recommends acceptance. I too am satisfied with the changes that you and your team made, and believe that you have adequately addressed all of the changes recommended by the Reviewer and the editors.

I am recommending that the journal accept the revised manuscript as is for publication in Extinction.

Sincerely

Bill F.